# Volume-Outcome Relationship in Surgical and Cardiac Transcatheter Interventions with a Focus on Transcatheter Aortic Valve Implantation

**DOI:** 10.3390/jcm11133806

**Published:** 2022-06-30

**Authors:** Sarah Mauler-Wittwer, Stephane Noble

**Affiliations:** Structural Cardiology Unit, University Hospital of Geneva, 4 Rue Gabrielle-Perret-Gentil, 1205 Geneva, Switzerland; sarah.mauler-wittwer@hcuge.ch

**Keywords:** transcatheter aortic valve implantation volume-outcome relationship, cost, complications, failure to rescue

## Abstract

“Practice makes perfect” is an old saying that can be true for complex interventions. There is a strong and persistent relationship between high volume and better outcomes with more than 300 studies being reported on the subject. The more complex the procedure, the greater the volume-outcome relationship is. Failure to rescue was shown to be one of the factors explaining higher mortality rates post complex surgery. High-volume centers provide a better safety net, thanks to the structure and better protocols, and low-volume operators have better results at high-volume centers than at low-volume centers. Finally, effort should be made to regroup complex procedures in high-volume centers, but without compromising patient access to the procedures. Adaptation to local and geographic constraints is important.

## 1. Introduction

“Practice makes perfect” is an old saying that can be true for complex interventions. There is a strong and persistent relationship between high volume and better outcomes with more than 300 studies being reported on the subject [1]. In this review article, we analyzed the correlation between the volume of interventions and their outcome for complex cardiac interventions, percutaneous coronary interventions (PCI) and valvular therapies with a focus on transcatheter aortic valve implantation (TAVI). The literature review was performed by searching for procedural volume and outcome in cardiovascular surgery and TAVR and references from major articles were also assessed as well as position papers and guidelines.

Procedural outcome depends as much on the available resources at the hospital as how well the surgeon performs the intervention. Patients treated by low-volume surgeons have higher mortality rates independently of the hospital volumes. In a study based on administrative data in which mortality was assessed for eight cardiovascular interventions or cancer resection between 1998–1999 in the United States, the strongest association with surgeon volume was reported for surgical aortic valve replacement (SAVR) (adjusted operative mortality: 9.1% when surgeon volume is <22 annual interventions and 6.5% when >42, 7.8% between 22 and 42) and repair of an abdominal aortic aneurysm (AAA) (adjusted operative mortality: 6.2% when operator volume is <8 cases per year, 4.6% between 8 and 17.5, and 3.9% when >17.5) [2]. Indeed, for these interventions, not only the technical skill of the surgeon but also the intra-operative process are potentially important elements for the outcome.

Interestingly, in lung resection in which patients rarely die from technical complications during the intervention itself (i.e., bleeding or air leakage from a bronchial stump), surgeon volume seems less important than hospital volume. Following a lung resection, the perioperative care in the intensive care unit (ICU) as well as the respiratory and nursing care are essential and thus hospital-based services are very important, especially considering that the hospital stay can be relatively long. At the other extreme, carotid endarterectomy requires a very short hospital stay without ICU admission and the intervention success relies more on the operative technique (e.g., intra-arterial shunt insertion and delicate patch angioplasty), thus the surgeon volume is more important than the hospital volume.

Importantly, in this study the mortality rate decreased with increasing operator volume but was not substantially modified by hospital volume when assessing carotid endarterectomy and SAVR. On the other hand, the adjusted mortality rate for lung resection was strongly related to hospital volume but less to operator volume [2].

In many interventions, the technical skill of the operator is important to prevent complications (e.g., bleeding or tissue devascularization) and may be associated with shorter operating time and better overall results. In 2013, in Michigan state, the relationship between the surgical skill of 20 bariatric surgeons and their risk-adjusted complication rates were assessed [3]. Ten peer surgeons anonymously performed a blinded assessment and graded each of the 20 voluntary surgeons by viewing a videotaped laparoscopic gastric bypass they had performed. Five domains of technical skill were assessed: gentleness, tissue exposure, instrument handling, time and motion, and flow of the operation. The mean grades across the 20 surgeons ranged from 2.6 to 4.8 (5: highest grade corresponding to the skill of a master bariatric surgeon, 3: average practicing bariatric surgeon and 1: skill of a general-surgery chief resident). Greater skills were associated with fewer peri-procedural complications (lowest quartile of surgical skill 14.5% versus highest quartile 5.2%, *p* < 0.001), lower mortality rate (0.26% versus 0.05%, *p* < 0.001), shorter operations (137 min versus 98 min, *p* < 0.001), lower rate of reoperation (3.4% vs. 1.6%, *p* < 0.001) and readmission (6.3% vs. 2.7%, *p* < 0.001). As a consequence, assessing surgeon skills using recorded procedures might be an avenue to explore when renewing certification and could potentially help surgeons improve or correct their techniques.

The volume-outcome relationship in the modern era was reported using national Medicare claims data from 2000 to 2009 which involved more than three million patients who underwent one of eight gastrointestinal, cardiac and vascular procedures [4]. Despite improvement in surgical safety, the strong relationship between higher volume and lower mortality still exists in recent years. Nevertheless, the authors suggest that volume should not be an exclusive measure of surgical quality. For rare operations, volume plays an important role in favoring outcomes whereas for frequently performed procedures direct measurements of minor and major complications and functional outcomes are important appropriate evidence of the surgical quality of a center.

Failure to rescue was shown to be one of the factors explaining higher mortality rates post complex visceral surgery in low-volume compared to high-volume hospitals [5]. Indeed, despite similar major complication rates between high- and low-volume hospitals, the chance to survive a complication was two–three times higher in a high-volume hospital. Patient-level data on almost 120,000 Medicare beneficiaries undergoing AVR, CABG and AAA repair between 2005 and 2006 showed that hospital volume was related more to failure to rescue rates than to complication rates [6]. The median procedure volumes at the lowest and highest volume hospitals were respectively 87 and 591 for CABG, 27 and 274 for AVR, 14 and 169 for AAA repair. The lowest-volume hospitals had significantly higher risk-adjusted mortality rates for the three different cardiovascular operations. Major post-operative complications were also significantly higher in low-volume hospital for AVR (OR: 1.12, 95% CI 1.06–1.18) and AAA repair (OR: 1.18, 95% CI 1.09–1.27). Indeed, failure to rescue was higher at lowest volume hospitals as shown by higher mortality rates after serious complications [6].

## 2. Cardiac Interventions

In the field of heart transplantation, several studies have shown that the one-year outcome was better in high-volume centers [7,8]. The analysis of >13,000 heart transplantations in 147 US centers between 1999 and 2005 showed that donor and recipient risk-adjusted one-year survival was better in centers performing a higher volume of heart transplantations [7]. Similarly, recipient patients had higher one-year mortality when treated at low-volume centers (<7 annual heart transplants) compared to high-volume centers (>15 annual procedures) (OR: 1.58; 95% CI: 1.30–1.92; *p* < 0.001) [8].

In the field of PCI, a cohort study of 60,000 patients from New York state between 1991 and 1994 showed an inverse relationship between hospital as well as operator volumes and mortality after PCI [9]. At that time, PCI was often limited to balloon angioplasty with worse outcomes than in the stent era. Nevertheless, patients treated by PCI in hospitals with annual volume <600 procedures and by operators performing <75 PCI had significantly higher mortality rates [9].

In the early 2000s, the Leapfrog group, a consortium of large corporations and public agencies that purchase healthcare, was an advocate of the importance of volume-outcome relationships and recommended that payers contract with hospitals with annual volume of >400 annual PCI [10]. Similarly, in 2001 the American College of Cardiology (ACC) recommended that PCI centers and operators should perform >400 and >75 annual procedures, respectively [11].

A report in the stent era on the volume-outcome relationship in New York state between 1998 and 2000 showed that the odds ratio for low-volume hospitals (<400 procedures) vs. high-volume hospitals was 1.98 for in-hospital mortality. The operator-volume threshold with the best odds ratio was 75 annual procedures and the odds ratio for low-volume operators was 1.3 for in-hospital mortality [12].

In 2013, the ACC reduced the minimum number of required PCI performed annually by each operator from 75 to 50, averaged over 2 years [13]. In Switzerland, the latest recommendations from 2014 suggest a minimum of 50 PCI per operator for elective cases and 75 in the cases of STEMI (ST-elevation myocardial infarction) treatment with a minimal global volume of 300 PCI in a center performing primary PCI [14].

Using the National Cardiovascular Data Registry involving 3,747,866 PCI performed by 10,496 operators between 2009 and 2015, the volume-outcome relationship was analyzed in the drug eluting stent era [15]. The median annual PCI volume per operator was 59 (IQR: 21–106 PCI) and 44% of the operators performed fewer than the recommended number of 50 PCI per year whereas 29% performed 50 to 100 PCI and 27% performed more than 100 PCI annually. Low-volume operators had higher mortality rates and more post PCI acute kidney injury when they performed PCI in low-volume centers. Indeed, high-volume centers provide a better safety net, thanks to the structure and better protocols.

Interestingly, the low-volume operators performed more primary PCI, but fewer complex PCI, less frequently treated multiple lesions in the same session, and their patients had fewer comorbidities than the intermediate- and high-volume operators. The modest increase in hospital mortality when patients with STEMI are treated by low-volume operators suggests that these low-volume operators are important to maintain access to primary PCI especially in rural and remote area [15].

Mortality rate is low when treating a simple coronary lesion, compared to calcified complex lesions requiring rotational atherectomy or recanalization of chronic total occlusion (CTO) or treatment of left main stem. Therefore, mortality may no longer be the appropriate measure for quality and volume assessment. The annual PCI number could also be put into perspective with lifetime experience of the operators and the complexity of the treated lesions. A too severe regulation of the number of required PCI may reduce PCI access for some patients without improving outcomes and may encourage the operators to perform borderline indicated PCI to reach the recommended number.

With respect to complex procedures involving rotational atherectomy, which is technically challenging and associated with procedural complications (e.g., coronary perforation, coronary dissection, slow flow) that are more frequent than after simple PCI, the analysis of 133,970 PCI with 7740 rotational atherectomy from the British national PCI database (2013–2016) showed no association between PCI volume and 30-day mortality on all PCI [16]. However, major adverse cardiovascular events increased after rotational atherectomy when the operator had performed <4 per year. Importantly, 55% of the operators did not reach this target volume and the median number of annual procedures was 2.5 (range: 0.25–55.25) [16]. Therefore, volume seems important in improving the outcome post rotational atherectomy. Other studies also showed a relationship between volume and outcome in complex procedures [17,18].

The British national PCI database also provided data on the volume-outcome relationship in 6724 unprotected left main PCI between 2012 and 2014 [18]. The median number of procedures per operator was thre3 per year. The operator volume ranged from 1 to 54 per year and 347 operators performed a median of two annual procedures, 134 performed a median of five, 59 performed a mean of 10 and 29 performed a mean of 21 annual left main PCI. In-hospital major cardiac and cerebrovascular events were lower and 12-months survival was better in the highest-volume operator group (mean of 21 annual procedures) compared to the lowest-volume operator group (median of two annual procedures). The authors estimated that the threshold to have better outcomes after unprotected left main stenting was 16 annual procedures [18].

Finally, with respect to recanalization of CTO, which is a challenging procedure with a success rate ranging from 45% to 90%, the volume-outcome relationship was reported using the 2010–2018 data from the Blue Cross Blue Shield of Michigan Cardiovascular Consortium registry [17]. Among the 210,172 patients included in the registry from 46 centers, 3.4% (7389) benefited from an attempt of recanalization with a 53% success rate. Success rates increased from 45% to 65% with operator experience and was the highest for high-volume operators (>33) at high-volume centers and the lowest for low-volume operators (<12) at low-volume centers. Low-volume operators had better results at high-volume centers than at low-volume centers. Table 1 summarizes the studies assessing the volume-outcome relationship for non-valvular procedures.

## 3. In the Field of Valvular Therapies

Before the TAVI era, SAVR was the most commonly performed valve surgery. The Michigan Society of Thoracic and Cardiovascular Surgeons Quality Collaborative, a multi-disciplinary group including 33 hospitals performing cardiac surgery in Michigan State, analyzed the volume-outcome relationship in a cohort of >6000 SAVR between 2008 and 2011 [19]. They used population-based data with robust clinical data. Hospital volume was identified as an important predictor of mortality. A logistic regression analysis using increments of 20 SAVR over a year showed that hospital volume was an independent predictor of risk-adjusted mortality. Furthermore, beyond a volume of 390 cases over 4 years, all hospitals had an observed/expected mortality ratio <1. In this study, hospital volumes but not operator volumes were an independent risk factor for early mortality. Indeed, this is in contrast to another study, in which operator volumes were critical for the outcome of SAVR [2].

Using observational data, the evolution of SAVR volume was assessed in the United States in the context of the advent and FDA (Food and Drug Administration) approval of TAVI for prohibitive surgical risk patients in 2011 and high surgical risk patients in 2012 [20]. Between 2008 and 2013, overall SAVR volumes have slightly increased (22%) since FDA approval of TAVI with a greater rise at hospitals offering TAVI. Increase of volume was more modest (16%) in non-TAVI centers. Furthermore, in-hospital mortality rate and risk-adjusted mortality for patients treated by SAVR were lower since the introduction of TAVI, particularly for patients at high surgical risk, who were the initial target population for TAVI. Interestingly, when comparing years before and after commercial availability of TAVI, the observed/expected ratio for in-hospital mortality after TAVI was higher at new TAVI centers but not at established centers. The explanation could be related to new operators facing their learning curve.

The effect of hospital volume on TAVI outcome was assessed using the National Inpatient Sample database of 2012 involving 1481 TAVI out of 7405 TAVI performed in 250 centers. They excluded patients with missing data on age, gender or death and patients younger than 60 [21]. The in-hospital mortality rate was lower at high-volume centers (first quartile (lowest volume): 6.4%, second quartile: 5.9%, third quartile: 5.2%, fourth quartile (highest volume): 2.8%) as well as complication rates (first quartile 48.5%, second quartile 44.2%, third quartile 39.7%, fourth quartile 41.5%). Furthermore, increasing hospital volume was associated with shorter length of stay and lower costs.

Similarly, in-hospital outcomes after TAVI were assessed according to hospital volume using the same National Inpatient Sample database of 2012 involving 7660 patients treated by TAVI in 256 centers [22]. The volume per center ranged from 5 to 255 cases with a median and mean of 20 and 29.8 cases, respectively. Of note, some low-volume centers were probably establishing their program. Using multivariate logistic regression analysis, a low-volume hospital was an independent predictor of death and bleeding [22].

Outside of the United States, the TAVI volume-outcome relationship was reported using the national quality registry data from 87 German hospitals with a total of 9924 patients treated in 2014 [23]. Among the 87 hospitals, 53% had a TAVI volume >100 trans-femoral procedures representing 79% of the total TAVI number. Finally, only 16% of the centers performed >200 TAVI procedures and the spectrum of hospital volumes ranged from 11 to 415 trans-femoral procedures annually with a continuous statistically significant association of lower average observed as well as risk-adjusted in-hospital mortality with increasing trans-femoral TAVI volumes. The in-hospital mortality was 5.6 ± 5.0% (range 0–16.7%) in the centers performing <50 annual trans-femoral TAVI compared to 2.4 ± 1.0% (range 0.5 to 3.7%) in the centers with >200 annual trans-femoral TAVI. In relation to increasing hospital volume, there was a significant trend towards low observed/expected ratios of in-hospital mortality. In the centers with <50 annual trans-femoral TAVI, the observed/expected ratio was 1.1 ± 1.0 (range: 0 to 3.9) compared to 0.5 ± 0.2 (range: 0.1 to 0.7) in the centers with a volume of >200 annual procedures [23]. Importantly, none of the high-volume hospitals (>200 annual procedures) had an observed/expected ratio > 1, whereas 41% (9 of 22) of low-volume hospitals (<50 annual procedures), 53% (10/19) of hospitals with 50 to 99 annual procedures, 28% (7/25) with 100 to 149 procedures and 29% (2/7) with 150 to 199 procedures had an observed/expected ratio > 1.

The authors concluded that the average in-hospital mortality was decreased by a factor of two in the highest volume centers (>200 annual procedures) in comparison with low-volume centers (<100 annual cases) and intermediate-volume centers (100–200 procedures). Their data suggested better average observed and risk-adjusted in-hospital mortality with increasing TAVI volumes [23]. On top of the in-hospital outcomes, we can also expect shorter hospital length of stay and lower cost in correlation with an increased volume of procedure [21].

More recent data on the volume-outcome relationship were reported using the TVT (Transcatheter Valve Therapy) registry from 2015 to 2017 with 113662 TAVI procedures performed by trans-femoral approach in 84.7% of the cases at 555 hospitals by 2960 operators [24]. Again, the authors found a significant inverse association between volume of trans-femoral TAVI and mortality, this time at 30 days post procedure, which remains after exclusion of the patients treated during the first 12-month start-up period at each center. The median annual volume per center was 54 (interquartile range: 36–86) and per operator was 27 (interquartile range: 17–43). Only seven centers performed >250 annual cases and only 200 operators performed >75 annual cases. The difference in adjusted mortality between the low-volume centers (mean volume: 27 annual procedures) and the high-volume centers (mean volume: 143 annual procedures) was a relative reduction of 19.45% (95%CI: 8.63–30.26). The adjusted 30-day mortality was 3.19% for the low-volume centers and 2.66% for the high-volume centers. There was a wide variation in hospital mortality in the low-volume centers. There was also a nonlinear association between unadjusted and adjusted mortality and operator annual volume. The difference in adjusted mortality between the low-volume operators (mean volume: 11 annual procedures) and the high-volume operators (mean volume: 70 annual procedures) was a relative reduction of 24.25% (95%CI: 10.40–38.10). The adjusted 30-day mortality was 3.54% for the low-volume operators and 2.84% for the high-volume operators. With respect to 30-day complications, there was no association with center volumes, except for the outcome of major vascular complications and major bleeding (10.03% in low-volume centers and 8.21% in high-volume centers) (OR: 1.25; 95%CI: 1.08–1.45) [24].

In 2021, a meta-analysis reported the effect of center volume for TAVI and mortality [25]. Seven studies met the criteria to be included in the meta-analysis which involved 193,498 TAVI (trans-apical approach: 1.3%) performed across the world. Centers were categorized as low (30–50 cases), intermediate (51–74 cases) or high (75–130 cases) annual volume, with respectively 25,062, 77,093 and 91,343 TAVI performed in these different categories. The absolute all-cause mortality rates were 5.15%, 3.66%, 3.24% in the low-, intermediate- and high-volume centers, respectively. The authors showed a relative reduction in mortality rates of 37% for high- compared to low-volume centers, of 23% for high- compared to intermediate-volume centers and of 19% for intermediate- compared to low-volume centers [25]. There were no differences in major vascular complications, but there was a trend for fewer major bleeding events at high-(6.0%) and intermediate-volume (7.5%) centers compared to low-volume centers (10.1%).

As described, several different studies have shown a volume-outcome relationship when performing TAVI. There was only one study which did not confirm this relationship when using rigorous multivariate regression analysis. It assessed 7365 TAVI procedures from the National Inpatient Sample in 2016, the largest publicly available all-payer inpatient care database in the United States [26]. In hospitals performing between 20 and 39 annual TAVI procedures, in-hospital mortality was nearly twice the mortality reported in the highest-volume centers (7.0 vs. 3.6%, *p* = 0.023). However, rigorous multivariate regression analysis did not show such volume-outcome relationship. Nevertheless, this study has some limitations such as the fact that the database relies on ICD-9 codes, which are not rigorously defined and that they did not use the VARC (Valve Academic Research Consortium) definitions. Table 2 summarizes the studies assessing the volume-outcome relationship for aortic valve replacement.

In 2017, the degree to which increasing experience is associated with outcome was explored using data from >42,000 procedures performed at 395 centers [28]. They found that increasing center volume was associated with lower in-hospital risk-adjusted outcome, namely mortality, vascular complications and bleeding but not stroke. Risk-adjusted adverse outcome (mortality, bleeding, vascular complications and stroke) declined from the first to four hundredth cases [28]. There was a higher rate of vascular complications and bleeding for the first hundred cases. The association between increasing case volume and lower in-hospital mortality, vascular complications and bleeding was most pronounced during the first hundred cases, corresponding to an early learning curve. After the first hundred cases, the risks associated with TAVI continued to decline but more gradually [28].

Similarly in 2018, Wassef et al. tried to determine the procedural learning curve of TAVI and the minimum annual center volumes associated with optimal TAVI outcomes by analyzing the international TAVR registry involving 16 centers around the world and 3403 patients [29] (Table 3). TAVI centers were categorized into low-volume (<50 cases annually), moderate-volume (50 to 100), and high-volume (>100) centers. Operator experience was also categorized into initial (1–75), early (76–150), intermediate (151–225), high (226–300) and very high (>300) experience in order to characterize the learning curve. All-cause mortality consistently decreased with increased experience in unadjusted analyses. Death, major vascular complications and major bleeding as well as early safety endpoints were all at the lowest when the TAVI was performed by a very experienced operator (>300 procedures). An experience of >225 TAVI was associated with lower 30-day-mortality whereas the early safety endpoints continued to improve beyond an experience of 225 procedures. Furthermore, the results of the multivariate regression analysis showed that TAVI volume was independently associated with 30-day mortality, major bleeding and major adverse cardiac events when used as a continuous variable.

Low-volume TAVI centers (<50 annual procedures) are independently associated with higher mortality. Major bleeding and early safety endpoints are also at the highest when TAVI are performed at low-volume centers. Adequate experience and training are essential to optimize the outcomes after TAVI. Furthermore nowadays, the absence of mortality is not enough to consider a successful TAVI. In 2022, a successful procedure means a patient alive, with no stroke, no para-valvular leak, a mean gradient <20 mmHg, no need for permanent pacemaker implantation and no major bleeding or vascular complications.

In 2019, the relationship between individual operator experience and TAVI outcome was assessed in a cohort of 8771 TAVI in New York performed between 2012 and 2016 by 207 operators [27]. One-third of the centers performed annually <83 annual cases and one-third of the operators performed <24 annual procedures. Results from high-volume operators (>80 annual procedures) showed a significantly lower risk of death, stroke or acute myocardial infarction compared with low-volume operators (<24 annual procedures) after adjusting for patient demographics as well as hospital and physician characteristics. Operators performing >200 procedures annually exposed their patients to a lower risk of post-procedural stroke and of a composite endpoint (i.e., mortality, stroke or myocardial infarction). The association between operator volume and outcome was most pronounced for the first 20 cases. Beyond 20 cases, a more gradual and linear improvement in risk-adjusted outcomes was associated with increasing operator volume. Indeed, mortality was not strongly related to operator volume. The difference in the composite endpoints relied mostly on stroke. Precise and direct optimal positioning of the trans-catheter heart valve with avoidance of recapture or second valve implantation contribute to reduced procedural stroke. Appropriate valve selection is another skill developed with experience that can reduce complications.

Numerous efforts have been made to reduce the learning curve such as proctoring at the beginning of the experience and continuous support by the specialist technician from the device industries.

Recommendation from the different medical societies is to start and maintain a TAVI program.

In Germany, a minimum hospital volume of 50 annual TAVI procedures was recommended in 2016 by the German Cardiac Society to guarantee appropriate standard and quality of care [21,23].

A consensus document from 2012 in the United States recommended the introduction of TAVI in centers performing >1000 catheterizations and 400 PCI annually and who also perform an annual volume of 50 SAVR [30]. Each center needs at least two cardiac surgeons on site who should have performed >100 SAVR with at least 10 considered as high risk. According to the document, cardiologists performing TAVI should have performed 100 left-sided structural procedures in their career or at least 30 left-sided structural procedures per year.

The consensus paper from 2018 recommended a TAVI program perform 50 cases annually or 100 cases over 2 years and registry-reported 30-day risk adjusted all-cause mortality, neurological events, vascular complications and major bleeding above the bottom 10%. The center should perform 300 PCI and 30 SAVR annually or 60 over 2 years with two hospital-based cardiac surgeons [31]. These recommendations not only considered volume, but also insist on the availability of a multi-disciplinary team and assessment of the procedural outcome. The operators should be able to analyze the aortic root, the annulus and the vascular access on a CT scan.

The Canadian Society of Cardiology published similar recommendations for operators to perform TAVI (participating in 100 TAVI procedures, 50 as first operators, CT scan training dedicated for TAVI) and for a center to perform TAVI it should perform 50 procedures annually [32].

However, the latest NCD (national coverage determination) and CMS (Centers for Medicare and Medicaid Services) from June 2019 are similar to the 2012 recommendations with respect to the criteria to maintain a TAVI program: 50 AVR (TAVI/SAVR) annually or 100 over a 2-year period, including 20 TAVI annually or 40 over two years. The criteria to establish a new TAVI program were also decreased to a volume of 20 SAVR in the two years before starting the program [25].

These new criteria are not in agreement with the medical literature showing a clear volume-outcome relationship. However, if more strict criteria were implemented, then up to 40% of the centers would no longer be able to perform TAVI in the United States. With the extension of the indication to low-risk patients, the number of TAVI is expected to increase and reducing the number of centers could eventually limit access in such a vast country as the United States. However, this argument does not apply in small European countries such as Switzerland, where indeed no recommendations are available.

## 4. TAVI Program without Cardiac Surgery

Until now the European and American guidelines have strongly recommended only performing TAVI in centers with on-site cardiac surgery [33,34]. The rate of conversion to surgery, however, is rather low, approximately 0.6% for trans-femoral approach and 1 to 1.5% overall. The associated mortality with conversion to surgery is around 45% at 30 days according to data from registries assessing first- and second-generation devices. Over the last few years, the incidence of surgical bailout has decreased with the use of new devices, which can often be recaptured and repositioned, as well as the increasing experience of the operators.

In some areas of the world, there are significant waiting lists (3 months, up to 5 to 8 months in the United Kingdom (UK) depending on the geographic area) associated with mortality (4.9% in Israel up to 23% in the UK) while waiting for the intervention [35]. A reason to perform TAVI without on-site cardiac surgery would be to ensure access to the therapy and equality of treatment in the rural areas. This is definitely not a need in Switzerland.

Nevertheless, we could image that high volume operators and their teams could have similar or better outcomes without on-site cardiac surgery than a low-volume center with on-site cardiac surgery. In our opinion, TAVI should be performed by adequately trained teams with sufficient yearly procedural volume and demonstration of outcomes similar to national benchmarks. A way to make sure that these conditions are fulfilled in countries without geographic constraints or long waiting lists is to recommend only performing TAVI in centers with on-site cardiac surgery. However, in a country such as the UK, where the penetration of TAVI is low, the population may benefit from the opening of centers without one-site cardiac surgery, since the need for bailout surgery is low and the mortality rate on the waiting list (2–3% per month) is high [35].

In conclusion, the more a procedure is carried out, the better one becomes at it, with as a consequence a better result. The more complex the procedure, the greater the volume-outcome relationship is. Failure to rescue was shown to be one of the factors explaining higher mortality rates post complex surgery. High-volume centers provide a better safety net, thanks to the structure and better protocols and low-volume operators have better results at high-volume centers than at low-volume centers. Finally, effort should be made to regroup complex procedures in high-volume centers, but without compromising patient access to the procedures. Adaptation to local and geographic constraints is important.

## Figures and Tables

**Table 1 jcm-11-03806-t001:** Studies assessing the volume-outcome relationship for non-valvular procedures.

	Number of Patients	Year	Location	Results
Bariatric surgery [3]	Assessment of the relationship between the skills of 20 surgeons and their risk-adjusted complication rates on 10,343 patients	2006–2012	Michigan state	Greater skills were associated with: -fewer peri-procedural complications (lowest quartile of surgical skill 14.5% vs. highest quartile 5.2%, *p* < 0.001) -lower mortality rate (0.26% vs. 0.05%, *p* < 0.001) -shorter operations (137 min vs. 98 min, *p* < 0.001) -lower rate of reoperation (3.4% vs. 1.6%, *p* < 0.001) -lower rate of readmission (6.3% vs. 2.7%, *p* < 0.001)
AVR, CABG and AAA [6]	120,000 Medicare beneficiaries	2005–2006	USA	Hospital volume was related more to failure to rescue rates than to complication rates
heart transplantation [7]	13,000	1999–2005	147 US centers	Donor and recipient risk-adjusted 1-year survival was better in centers performing a higher volume of heart transplantations
PCI [9]	62,670	1991–1994	New York state	Patients treated by PCI in hospitals with annual volume <600 procedures and by operators performing <75 PCI had significantly higher mortality rates
PCI [12]	107,713	1998–2000	New York state	-The odds ratio for low-volume hospitals (<400 procedures) vs. high-volume hospitals was 1.98 for in-hospital mortality. -The operator-volume threshold with the best odds ratio was 75 annual procedures and the odds ratio for low-volume operators was 1.3 for in-hospital mortality
PCI [15]	374,7866 patients 10,496 operators	2009–2015	National Cardiovascular Data Registry USA	Low-volume operators had higher mortality rates and more post PCI acute kidney injury when they performed PCI in low-volume centers.
PCI with rotational atherectomy [16]	133,970 PCI with 7740 rotational atherectomy	2013–2016	British national PCI database	-No association between PCI volume and 30-day mortality on all PCI -Major adverse cardiovascular events increased after rotational atherectomy when the operator had performed <4 per year
Chronic total occlusion [17]	210,172 patients included in the registry from 46 centers, 7389 (3.4%) had recanalization attempt	2010–2018	Michigan	-Success rates increased from 45% to 65% with operator experience and was the highest for high-volume operators (>33) at high-volume centers and the lowest for low-volume operators (<12) at low-volume centers. -Low-volume operators had better results at high-volume centers than at low-volume centers
Unprotected left main PCI [18]	6724	2012–2014	British national PCI database	-In-hospital major cardiac and cerebrovascular events were lower and 12-months survival was better in the highest-volume operator group (mean of 21 annual procedures) compared to the lowest-volume operator group (median of 2 annual procedures). -The estimated threshold to have better outcomes after unprotected left main stenting was 16 annual procedures

AVR: Aortic valve replacement, CABG: coronary artery bypass graft, AAA: abdominal aortic aneurysm, PCI: percutaneous coronary artery disease.

**Table 2 jcm-11-03806-t002:** Studies assessing the volume-outcome relationship for aortic valve replacement.

	Number of Patients	Year	Location	Results
8 cardiovascular interventions or cancer resection [3]	474,108 in total, (for SAVR: NA)	1998–1999	USA	Adjusted operative mortality for SAVR: 9.1% when annual surgeon volume is <22 7.8% when between 22 and 42 6.5% when >42
SAVR [19]	6270	2008–2011	Michigan State	-Hospital volumes but not operator volumes were an independent risk factor for early mortality -Beyond a volume of 390 cases over 4 years, all hospitals had an observed/expected mortality ratio <1
TAVI [21]	1481 TAVI out of 7405 TAVI performed in 250 centers	2012	National Inpatient Sample database of 2012	-The in-hospital mortality rate was lower at high-volume centers (first quartile: 6.4%, second quartile: 5.9%, third quartile: 5.2%, fourth quartile: 2.8%) as well as complication rates (first quartile: 48.5%, second quartile 44.2%, third quartile 39.7%, fourth quartile 41.5%). -Increasing hospital volume was associated with shorter length of stay and lower costs.
TAVI [23]	9924	2014	Germany (87 hospitals)	-The in-hospital mortality was 5.6 ± 5.0% (range 0–16.7%) in the centers performing <50 annual transfemoral TAVI compared to 2.4 ± 1.0% (range: 0.5 to 3.7%) in the centers with >200 annual transfemoral TAVI. -In relation to increasing hospital volume there was a significant trend towards low observed/expected ratios of in-hospital mortality
TAVI [24]	113,662	2015–2017	TVT registry	-Significant inverse association between volume of transfemoral TAVI and mortality -The median annual volume per center was 54 (IQR: 36–86) and per operator was 27 (IQR: 17–43) -The adjusted 30-day mortality was 3.19% for the low-volume centers and 2.66% for the high-volume centers.
TAVI [25]	193,498	2021	Meta-analysis	-The absolute all-cause mortality rates were 5.15%, 3.66%, 3.24% in the low-(30–50 cases), intermediate-(51–74) and high-volume (75–130) centers, respectively. -Relative reduction in mortality rates of 37% for high- compared to low-volume centers, of 23% for high-compared to intermediate-volume centers and of 19% for intermediate- compared to low-volume centers
TAVI [26]	7365	2016	National Inpatient Sample	-In hospitals performing between 20 and 39 annual TAVI procedures, in-hospital mortality was 7.0% compared to 3.6% in the highest-volume centers (*p* = 0.023) -However, rigorous multivariate regression analysis did not show such volume-outcome relationship.
TAVI [27]	8771	2012–2016	New York	-High-volume operators (>80 annual procedures) had a significantly lower risk of death, stroke or acute myocardial infarction compared with low-volume operators (<24) after adjusting for patient demographics as well as hospital and physician characteristics -Operators performing >200 procedures annually exposed their patients to a lower risk of post-procedural stroke and of a composite endpoint (i.e., mortality, stroke or myocardial infarction) -The difference in the composite endpoints relied mostly on stroke

The procedural learning curve of TAVI.

**Table 3 jcm-11-03806-t003:** Studies assessing the learning curve for TAVI.

Number of Patients	Location	Results
42,000 [28]	395 US centers	-Risk-adjusted adverse outcome (mortality, bleeding, vascular complications and stroke) declined from the first to 400 cases -Higher rate of vascular complications and bleeding for the first 100 cases -The association between increasing case volume and lower in-hospital mortality, vascular complications and bleeding was most pronounced during the first 100 cases -After the first 100 cases, the risks associated with TAVI continued to decline but more gradually
3403 [29]	16 centers in the world	-Death, major vascular complications and major bleeding as well as early safety endpoints were all at the lowest when the TAVI was performed by a very experienced operator (>300) -An experience of >225 TAVI was associated with lower 30-day-mortality whereas the early safety endpoints continued to improve beyond an experience of 225 procedures -Low-volume TAVI centers (<50 annual procedures) are independently associated with higher mortality -Major bleeding and early safety endpoints are also at the highest when TAVI are performed at low-volume centers

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
