# Peer review of "Volume-Outcome Relationship in Surgical and Cardiac Transcatheter Interventions with a Focus on Transcatheter Aortic Valve Implantation"

_jcm, 2022, doi:10.3390/jcm11133806_

Round 1

Reviewer 1 Report

The article is classified as a review, sometimes reading the article gives the impression that there are too many "personal impressions" not objectified by the data.

- The authors focus on the outcomes of interventional structural procedures, especially aortic ones.

- Is the history of PCI, one of the most performed and life-saving procedures in the world, exactly contrary to that of structural intervention? How can this discrepancy be explained?

- Speaking of opinion (according to the style of the article), what is the opinion regarding TAVI performed in centers without cardiac surgery?

- A summary chart on the data cited in the article would be useful

Author Response

The article is classified as a review, sometimes reading the article gives the impression that there are too many "personal impressions" not objectified by the data.

One of our objectives was to base our article on data rather than on personal impressions. We used 35 references and we searched for article on procedural volume and outcome in cardiovascular surgery and TAVR and references from the major articles were also assessed as well as position papers and guidelines.

The only time we really give our opinion is in the new paragraph on TAVI without on-site cardiac surgery and state it at the beginning of the sentence:

In our opinion, TAVI should be performed by adequately trained teams with sufficient yearly procedural volume and demonstration of outcomes similar to national benchmarks.

- The authors focus on the outcomes of interventional structural procedures, especially aortic ones.

Yes, that was one of our objectives

- Is the history of PCI, one of the most performed and life-saving procedures in the world, exactly contrary to that of structural intervention? How can this discrepancy be explained?

We respectfully disagree with the comment that PCI is exactly the opposite than structural heart intervention. There is also a clear volume outcome relationship with respect to PCI

We added table 2 which is summarizing the data on PCI

- Speaking of opinion (according to the style of the article), what is the opinion regarding TAVI performed in centers without cardiac surgery?

Until now the European and American guidelines have strongly recommended only performing TAVI in centers with on-site cardiac surgery. The rate of conversion to surgery, however, is rather low, approximately 0.6% for transfemoral approach and 1 to 1.5% overall. The associated mortality with conversion to surgery is around 45% at 30 days according to data from registries assessing first- and second-generation devices. Over the last few years, the incidence of surgical bailout has decreased with the use of new devices which can often be recaptured and repositioned as well as the increasing experience of the operators.

In some areas of the world, there are significant waiting lists (3 months, up to 5 to 8 months in the UK depending on the geographic area) associated with mortality (4.9% in Israel up to 23% in the UK) while waiting for the intervention. A reason to perform TAVI without on-site cardiac surgery would be to ensure access to the therapy and equality of treatment in the rural areas. This is definitely not needed in Switzerland.

Nevertheless, we could image that high volume operators and their teams could have similar or better outcomes without on-site cardiac surgery than a low-volume center with on-site cardiac surgery. In our opinion, TAVI should be performed by adequately trained teams with sufficient yearly procedural volume and demonstration of outcomes similar to national benchmarks. A way to make sure that these conditions are fulfilled in countries without geographic constraints or long waiting lists is to recommend only performing TAVI in centers with on-site cardiac surgery. However, in a country such as the UK, where the penetration of TAVI is low, the population may benefit from the opening of centers without one-site cardiac surgery, since the need for bailout surgery is low and the mortality rate on the waiting list (2-3% per month) is high.

A summary chart on the data cited in the article would be useful

We added 3 tables summarizing the main data from the studies

Table 1: Studies assessing the volume-outcome relationship for non-valvular procedures

Number of patients

year

location

Results

Bariatric surgery (3)

Assessment of the relationship between the skills of 20 surgeons and their risk-adjusted complication rates on 10343 patients

2006-2012

Michigan state

Greater skills were associated with:

-fewer peri-procedural complications (lowest quartile of surgical skill 14.5% vs highest quartile 5.2%, p<0.001)

-lower mortality rate (0.26% vs 0.05%, p<0.001)

-shorter operations (137min vs 98min, p<0.001)

-lower rate of reoperation (3.4% vs 1.6%, p<0.001)

-lower rate of readmission (6.3% vs 2.7%, p<0.001)

AVR, CABG and AAA (6)

120000 Medicare beneficiaries

2005-2006

USA

Hospital volume was related more to failure to rescue rates than to complication rates

heart transplantation (7)

13000

1999-2005

147 US centers

Donor and recipient risk-adjusted 1-year survival was better in centers performing a higher volume of heart transplantations

PCI (9)

62670

1991-1994

New York state

Patients treated by PCI in hospitals with annual volume <600 procedures and by operators performing <75 PCI had significantly higher mortality rates

PCI (12)

107713

1998-2000

New York state

-The odds ratio for low-volume hospitals (<400 procedures) vs high-volume hospitals was 1.98 for in-hospital mortality.

-The operator-volume threshold with the best odds ratio was 75 annual procedures and the odds ratio for low-volume operators was 1.3 for in-hospital mortality

PCI (15)

3747866 patients

10496 operators

2009-2015

National Cardiovascular Data Registry USA

Low-volume operators had higher mortality rates and more post PCI acute kidney injury when they performed PCI in low-volume centers.

PCI with rotational atherectomy (16)

133970 PCI with 7740 rotational atherectomy

2013-2016

British national PCI database

-No association between PCI volume and 30-day mortality on all PCI

-Major adverse cardiovascular events increased after rotational atherectomy when the operator had performed <4 per year

Chronic total occlusion (17)

210172 patients included in the registry from 46 centers, 7389 (3.4%) had recanalization attempt

2010-2018

Michigan

-Success rates increased from 45% to 65% with operator experience and was the highest for high-volume operators (>33) at high-volume centers and the lowest for low-volume operators (<12) at low-volume centers.

-Low-volume operators had better results at high-volume centers than at low-volume centers

Unprotected left main PCI (18)

6724

2012-2014

British national PCI database

-In-hospital major cardiac and cerebrovascular events were lower and 12-months survival was better in the highest-volume operator group (mean of 21 annual procedures) compared to the lowest-volume operator group (median of 2 annual procedures).

-The estimated threshold to have better outcomes after unprotected left main stenting was 16 annual procedures

Table 2: Studies assessing the volume-outcome relationship for aortic valve replacement

Number of patients

year

location

Results

8 cardiovascular interventions or cancer resection(2)

474 108 in total, (for SAVR: NA)

1998-1999

USA

Adjusted operative mortality for SAVR:

9.1% when annual surgeon volume is <22

7.8% when between 22 and 42

6.5% when >42

SAVR (19)

6270

2008-2011

Michigan State

-Hospital volumes but not operator volumes were an independent risk factor for early mortality

-Beyond a volume of 390 cases over 4 years, all hospitals had an observed/expected mortality ratio <1

TAVI (21)

1481 TAVI out of 7405 TAVI performed in 250 centers

2012

National Inpatient Sample database of 2012

-The in-hospital mortality rate was lower at high-volume centers (first quartile: 6.4%, second quartile: 5.9%, third quartile: 5.2%, fourth quartile :2.8%) as well as complication rates (first quartile: 48.5%, second quartile 44.2%, third quartile 39.7%, fourth quartile 41.5%).

-Increasing hospital volume was associated with shorter length of stay and lower costs.

TAVI (23)

9924

2014

Germany

(87 hospitals)

-The in-hospital mortality was 5.6±5.0% (range 0-16.7%) in the centers performing <50 annual transfemoral TAVI compared to 2.4±1.0% (range:0.5 to 3.7%) in the centers with >200 annual transfemoral TAVI.

-In relation to increasing hospital volume there was a significant trend towards low observed/expected ratios of in-hospital mortality

TAVI (24)

113662

2015-2017

TVT registry

-Significant inverse association between volume of transfemoral TAVI and mortality

-The median annual volume per center was 54 (IQR: 36-86) and per operator was 27 (IQR: 17-43)

-The adjusted 30-day mortality was 3.19% for the low-volume centers and 2.66% for the high-volume centers.

TAVI (25)

193498

2021

Meta-analysis

-The absolute all-cause mortality rates were 5.15%, 3.66%, 3.24% in the low-(30-50 cases), intermediate- (51-74) and high-volume (75-130) centers, respectively.

-Relative reduction in mortality rates of 37% for high- compared to low-volume centers, of 23% for high- compared to intermediate-volume centers and of 19% for intermediate- compared to low-volume centers

TAVI (26)

7365

2016

National Inpatient Sample

-In hospitals performing between 20 and 39 annual TAVI procedures, in-hospital mortality was 7.0% compared to 3.6% in the highest-volume centers (p=0.023)

-However, rigorous multivariate regression analysis did not show such volume-outcome relationship.

TAVI (29)

8771

2012-2016

New York

- High-volume operators (>80 annual procedures) had a significantly lower risk of death, stroke or acute myocardial infarction compared with low-volume operators (<24) after adjusting for patient demographics as well as hospital and physician characteristics

- Operators performing >200 procedures annually exposed their patients to a lower risk of post-procedural stroke and of a composite endpoint (i.e. mortality, stroke or myocardial infarction)

- The difference in the composite endpoints relied mostly on stroke

Table 3: Studies assessing the learning curve for TAVI

Number of patients

location

Results

42000 (27)

395 UScenters

-Risk-adjusted adverse outcome (mortality, bleeding, vascular complications and stroke) declined from the first to 400 cases

-Higher rate of vascular complications and bleeding for the first 100 cases

-The association between increasing case volume and lower in-hospital mortality, vascular complications and bleeding was most pronounced during the first 100 cases

- After the first 100 cases, the risks associated with TAVI continued to decline but more gradually

3403

(28)

16 centers in the world

-Death, major vascular complications and major bleeding as well as early safety endpoints were all at the lowest when the TAVI was performed by a very experienced operator (>300)

-An experience of >225 TAVI was associated with lower 30-day-mortality whereas the early safety endpoints continued to improve beyond an experience of 225 procedures

-Low-volume TAVI centers (<50 annual procedures) are independently associated with higher mortality

-Major bleeding and early safety endpoints are also at the highest when TAVI are performed at low-volume centers

Reviewer 2 Report

The authors reviewed the association between the learning curve of operators and adverse events with a focus on transcatheter aortic valve implantation, emphasizing the importance of high-volume centers. This paper is very interesting and well organized.

Major comments

1.     The reviewer has no major comment because the text is comprehensively described.

Minor comments

1.     The references are appropriate. However, no figures or tables are shown. The authors should add them.

2.     The number of characters per line is out of alignment. For example, the line up to 266 and the line from 268.

Author Response

The authors reviewed the association between the learning curve of operators and adverse events with a focus on transcatheter aortic valve implantation, emphasizing the importance of high-volume centers. This paper is very interesting and well organized.

 Major comments

  1. The reviewer has no major comment because the text is comprehensively described.

Thank you for your positive comment

Minor comments

  1. The references are appropriate. However, no figures or tables are shown. The authors should add them.

We added 3 tables summarizing the data

Table 1: Studies assessing the volume-outcome relationship for non-valvular procedures

Number of patients

year

location

Results

Bariatric surgery (3)

Assessment of the relationship between the skills of 20 surgeons and their risk-adjusted complication rates on 10343 patients

2006-2012

Michigan state

Greater skills were associated with:

-fewer peri-procedural complications (lowest quartile of surgical skill 14.5% vs highest quartile 5.2%, p<0.001)

-lower mortality rate (0.26% vs 0.05%, p<0.001)

-shorter operations (137min vs 98min, p<0.001)

-lower rate of reoperation (3.4% vs 1.6%, p<0.001)

-lower rate of readmission (6.3% vs 2.7%, p<0.001)

AVR, CABG and AAA (6)

120000 Medicare beneficiaries

2005-2006

USA

Hospital volume was related more to failure to rescue rates than to complication rates

heart transplantation (7)

13000

1999-2005

147 US centers

Donor and recipient risk-adjusted 1-year survival was better in centers performing a higher volume of heart transplantations

PCI (9)

62670

1991-1994

New York state

Patients treated by PCI in hospitals with annual volume <600 procedures and by operators performing <75 PCI had significantly higher mortality rates

PCI (12)

107713

1998-2000

New York state

-The odds ratio for low-volume hospitals (<400 procedures) vs high-volume hospitals was 1.98 for in-hospital mortality.

-The operator-volume threshold with the best odds ratio was 75 annual procedures and the odds ratio for low-volume operators was 1.3 for in-hospital mortality

PCI (15)

3747866 patients

10496 operators

2009-2015

National Cardiovascular Data Registry USA

Low-volume operators had higher mortality rates and more post PCI acute kidney injury when they performed PCI in low-volume centers.

PCI with rotational atherectomy (16)

133970 PCI with 7740 rotational atherectomy

2013-2016

British national PCI database

-No association between PCI volume and 30-day mortality on all PCI

-Major adverse cardiovascular events increased after rotational atherectomy when the operator had performed <4 per year

Chronic total occlusion (17)

210172 patients included in the registry from 46 centers, 7389 (3.4%) had recanalization attempt

2010-2018

Michigan

-Success rates increased from 45% to 65% with operator experience and was the highest for high-volume operators (>33) at high-volume centers and the lowest for low-volume operators (<12) at low-volume centers.

-Low-volume operators had better results at high-volume centers than at low-volume centers

Unprotected left main PCI (18)

6724

2012-2014

British national PCI database

-In-hospital major cardiac and cerebrovascular events were lower and 12-months survival was better in the highest-volume operator group (mean of 21 annual procedures) compared to the lowest-volume operator group (median of 2 annual procedures).

-The estimated threshold to have better outcomes after unprotected left main stenting was 16 annual procedures

Table 2: Studies assessing the volume-outcome relationship for aortic valve replacement

Number of patients

year

location

Results

8 cardiovascular interventions or cancer resection(2)

474 108 in total, (for SAVR: NA)

1998-1999

USA

Adjusted operative mortality for SAVR:

9.1% when annual surgeon volume is <22

7.8% when between 22 and 42

6.5% when >42

SAVR (19)

6270

2008-2011

Michigan State

-Hospital volumes but not operator volumes were an independent risk factor for early mortality

-Beyond a volume of 390 cases over 4 years, all hospitals had an observed/expected mortality ratio <1

TAVI (21)

1481 TAVI out of 7405 TAVI performed in 250 centers

2012

National Inpatient Sample database of 2012

-The in-hospital mortality rate was lower at high-volume centers (first quartile: 6.4%, second quartile: 5.9%, third quartile: 5.2%, fourth quartile :2.8%) as well as complication rates (first quartile: 48.5%, second quartile 44.2%, third quartile 39.7%, fourth quartile 41.5%).

-Increasing hospital volume was associated with shorter length of stay and lower costs.

TAVI (23)

9924

2014

Germany

(87 hospitals)

-The in-hospital mortality was 5.6±5.0% (range 0-16.7%) in the centers performing <50 annual transfemoral TAVI compared to 2.4±1.0% (range:0.5 to 3.7%) in the centers with >200 annual transfemoral TAVI.

-In relation to increasing hospital volume there was a significant trend towards low observed/expected ratios of in-hospital mortality

TAVI (24)

113662

2015-2017

TVT registry

-Significant inverse association between volume of transfemoral TAVI and mortality

-The median annual volume per center was 54 (IQR: 36-86) and per operator was 27 (IQR: 17-43)

-The adjusted 30-day mortality was 3.19% for the low-volume centers and 2.66% for the high-volume centers.

TAVI (25)

193498

2021

Meta-analysis

-The absolute all-cause mortality rates were 5.15%, 3.66%, 3.24% in the low-(30-50 cases), intermediate- (51-74) and high-volume (75-130) centers, respectively.

-Relative reduction in mortality rates of 37% for high- compared to low-volume centers, of 23% for high- compared to intermediate-volume centers and of 19% for intermediate- compared to low-volume centers

TAVI (26)

7365

2016

National Inpatient Sample

-In hospitals performing between 20 and 39 annual TAVI procedures, in-hospital mortality was 7.0% compared to 3.6% in the highest-volume centers (p=0.023)

-However, rigorous multivariate regression analysis did not show such volume-outcome relationship.

TAVI (29)

8771

2012-2016

New York

- High-volume operators (>80 annual procedures) had a significantly lower risk of death, stroke or acute myocardial infarction compared with low-volume operators (<24) after adjusting for patient demographics as well as hospital and physician characteristics

- Operators performing >200 procedures annually exposed their patients to a lower risk of post-procedural stroke and of a composite endpoint (i.e. mortality, stroke or myocardial infarction)

- The difference in the composite endpoints relied mostly on stroke

Table 3: Studies assessing the learning curve for TAVI

Number of patients

location

Results

42000 (27)

395 UScenters

-Risk-adjusted adverse outcome (mortality, bleeding, vascular complications and stroke) declined from the first to 400 cases

-Higher rate of vascular complications and bleeding for the first 100 cases

-The association between increasing case volume and lower in-hospital mortality, vascular complications and bleeding was most pronounced during the first 100 cases

- After the first 100 cases, the risks associated with TAVI continued to decline but more gradually

3403

(28)

16 centers in the world

-Death, major vascular complications and major bleeding as well as early safety endpoints were all at the lowest when the TAVI was performed by a very experienced operator (>300)

-An experience of >225 TAVI was associated with lower 30-day-mortality whereas the early safety endpoints continued to improve beyond an experience of 225 procedures

-Low-volume TAVI centers (<50 annual procedures) are independently associated with higher mortality

-Major bleeding and early safety endpoints are also at the highest when TAVI are performed at low-volume centers

  1. The number of characters per line is out of alignment. For example, the line up to 266 and the line from 268.

      We corrected

Reviewer 3 Report

Nice and interesting work, but the contribution to the field is average. Maybe you can estimate the average number of procedures which should be performed per year for best results, which could be helpful for guidelines development.

Author Response

Nice and interesting work, but the contribution to the field is average. Maybe you can estimate the average number of procedures which should be performed per year for best results, which could be helpful for guidelines development.

In our opinion the contribution to the field is to highlight the volume outcome relationship and provide some data to discuss the minimal number of cases required per center considering access to the therapy and the safety and efficacy of the procedure in different area. The Canadian Society of Cardiology published detailed recommendation in order to perform TAVI with a minimum of 50 TAVI per year. There is no recommendation in Switzerland. In Germany as well in the US, they also established recommendation. There is a tendency to consider structural heart interventions as a specific field of interventional cardiology requiring a dedicated training.

Reviewer 4 Report

This is an interesting review that presents data demonstrating a better outcome of procedures when performed in high-volume centers.

This is a well-known and widely accepted fact, but it is of interest to collect existing information specifically in the field of interventional cardiology and especially in TAVI, as done in the article.

Minor changes:

1) The objective (line 26) also includes complex non-cardiac interventions. This is too broad an approach, not met by the article. There are hundreds of publications on the subject and this review does not sufficiently cover the literature in the non-cardiac field. Although it makes sense to comment on key studies from other surgical/interventional fields in the introduction as a frame of reference, the objective (and the review itself) should be focused on interventional cardiac procedures only.

2) In the interventional cardiac field reviewed in detail, basically PCI and TAVI (and surgical  SAVR as a reference), even if this is not a systematic review, the authors should indicate how the sources to be considered for the review were selected and  briefly report the criteria for the literature search and selection. Otherwise there is a substantial risk of bias in their data selection and conclusions.

Author Response

This is an interesting review that presents data demonstrating a better outcome of procedures when performed in high-volume centers.

This is a well-known and widely accepted fact, but it is of interest to collect existing information specifically in the field of interventional cardiology and especially in TAVI, as done in the article.

Minor changes:

  • The objective (line 26) also includes complex non-cardiac interventions. This is too broad an approach, not met by the article. There are hundreds of publications on the subject and this review does not sufficiently cover the literature in the non-cardiac field. Although it makes sense to comment on key studies from other surgical/interventional fields in the introduction as a frame of reference, the objective (and the review itself) should be focused on interventional cardiac procedures only.

We agree with your comment and we changed the objective and kept the major studies about non cardiac interventions as frame of reference.

  • In the interventional cardiac field reviewed in detail, basically PCI and TAVI (and surgical SAVR as a reference), even if this is not a systematic review, the authors should indicate how the sources to be considered for the review were selected and briefly report the criteria for the literature search and selection. Otherwise, there is a substantial risk of bias in their data selection and conclusions.

Thank you for this comment and we added the following sentence

The literature review was performed by searching for procedural volume and outcome in cardiovascular surgery and TAVR and the references from the major articles were also assessed as well as position papers and guidelines.

Round 2

Reviewer 1 Report

Although my perplexities about this article persist, I must say that the authors have fully responded to my observations.